# A Novel Dentary Bone Conduction Device Equipped with Laser Communication in DSP

**DOI:** 10.3390/s21124229

**Published:** 2021-06-20

**Authors:** Jau-Woei Perng, Tung-Li Hsieh, Cheng-Yan Guo

**Affiliations:** 1Department of Mechanical and Electromechanical Engineering, National Sun Yat-Sen University, Kaohsiung 80424, Taiwan; jwperng@faculty.nsysu.edu.tw; 2General Education Center, College of Liberal Arts Education, Wenzao Ursuline University of Languages, Kaohsiung 80793, Taiwan; 3College of Medicine, National Taiwan University, Taipei 10617, Taiwan; r04458006@ntu.edu.tw

**Keywords:** DSP-based, CMU ARCTIC databases, DSP digital filter

## Abstract

In this study, we designed a dentary bone conduction system that transmits and receives audio by laser. The main objective of this research was to propose a complete hardware design method, including a laser audio transmitter and receiver and digital signal processor (DSP) based digital signal processing system. We also present a digital filter algorithm that can run on a DSP in real time. This experiment used the CMU ARCTIC databases’ human-voice reading audio as the standard audio. We used a piezoelectric sensor to measure the vibration signal of the bone conduction transducer (BCT) and separately calculated the signal-to-noise ratio (SNR) of the digitally filtered audio output and the unfiltered audio output using DSP. The SNR of the former was twice that of the latter, and the BCT output quality significantly improved. From the results, we can conclude that the dentary bone conduction system integrated with a DSP digital filter enhances sound quality.

## 1. Introduction

Sound is transmitted through bones. The auditory path begins with bone vibration and passes through the inner ear via the auditory nerve to the auditory center, without passing through the outer and middle ear. Sound is thus transmitted directly to the cochlea in the inner ear through the bone. This auditory path’s major function is to cause the cochlear wall to vibrate, allowing the inner ear sensors to receive the signal, which is then transmitted through the auditory nerve to the auditory center in the brain, which generates an auditory sensation. Figure 1 shows the structure of the human ear. People with hearing loss who have a normal auditory nerve can still utilize the bone conduction path to receive an auditory sensation. According to domestic and foreign research on bone conduction hearing aids, such a device usually includes a signal converter electrically connected to a vibrator. The signal converter receives the ambient sound signal and converts it into a corresponding electrical signal. A vibrating sheet is then driven by the electrical signal to generate a vibration. Users would simply press the vibrating sheet against the bones in their middle ear area to utilize the bone as a medium to transmit the frequency of the vibrating sheet to their cochlear structure and the auditory nerve, enabling perception of the sound received by the signal converter. However, the signal converter of a bone conduction hearing aid can receive sounds from all directions. Excessive sound sources or continual noise can prevent users from correctly receiving the target sound, resulting in low accuracy of sound transmission. Additionally, when an audio signal is transmitted through the air, its intensity attenuates with distance. Excessive distance between the user and the sound source can, thus, prevent smooth transmission of the audio signal to the signal converter, and the user will, therefore, be unable to receive sounds from distant locations, thus diminishing usability. Conventional bone conduction devices, therefore, require a redesign to resolve these defects of low accuracy and poor usability. The proposed transmission device employs dentary bone vibration for sound conduction. It receives an optical signal that also carries auditory information and limits users to receive only the audio information carried by the optical signal. This not only facilitates sound transmission accuracy but also improves usability by preventing audio signal attenuation with distance. In this research, we design a dentary bone conduction system that can transmit audio through a laser. After digitally filtering the laser signal through a digital signal processor (DSP), the filtered signal is output to the bone conduction transducer (BCT) in real time. Furthermore, the laser audio signal is converted into a mechanical wave transmitted to the dentary bone’s auditory system. Laser communication is widely used in aerospace engineering [1]. It operates at a low cost, requires low power consumption, and will not be disrupted by radio frequency communication during transmission [2]. In the laser communication system’s design, the transmitting end can receive the signal to be transmitted from the audio playback device and modulate the audio signal to the laser diode driver. The receiving end uses a photodiode (PD) or solar panel to receive the laser signal [3] and demodulate the received modulation signal to obtain the transmitted signal. Furthermore, the laser signal has the desirable characteristics of directivity and privacy [4].

Bone conduction research has made significant progress. Many commercial products, such as bone-anchored hearing aids (BAHAs) [5], have been developed through this research. BAHAs were used clinically in the 1970s through implantation [6]. However, the risk of complications from implant methods is relatively high [7]. In 2015, a commercial product literature review of bone conduction devices (BCDs) mentioned a non-implantable intraoral BCD [8] that transmits a piezoelectric transducer’s vibration signal to the skull through the teeth. A healthy cochlea can receive the vibration signal, which allows the patient to hear sound. According to clinical research results [9], an intraoral BCD does not cause adverse reactions or complications. Current non-implantable bone conduction research focuses on analyzing the transducer’s mechanical vibration signal [10] and the mathematical model of mechanical wave transmission in the skull [11]. A study of the transducer’s mechanical model shows that the transducer’s frequency can be described as the vibration of the mass and the damping and elastic coefficient of the system [12]. The study partly discusses the loss of mechanical wave transmission paths in soft tissue [13]. At present, there is little discussion of the application of DSP and digital signal processing methods in bone conduction system designs [14]. We used an intraoral BCD as the experimental platform for this article and describe a complete DSP realization process in addition to analyzing the dentary bone conduction system.

### Our Contribution

At present, the wearer of the bone conduction device will not hear their conversation partner’s voice on noisy occasions. That is because the microphone will receive the sound of the entire space and couple it. Our contribution is to propose and implement a dentary bone conduction system that can transmit voice through optical communication and a framework for processing the received optically transmitted voice signal through DSP. The schematic diagram of a dentary bone conduction system applied to humans is shown in Figure 2. The advantage of our proposed dentary bone conduction system compared with other types of bone conduction is that compared with the other types of bone conduction application situations, dental bone conduction does not need to transmit mechanical signals through the skin and soft tissues. Therefore, it does not cause mechanical signal attenuation, so the voice/sound can more clearly be heard. A laser can effectively avoid the transmission of voice signals of the cocktail party problem. The cocktail party effect refers to the brain’s ability for selective hearing. In this phenomenon, one’s auditory attention is focused on a person’s conversation while ignoring other conversations or noises in the background. This effect reveals an amazing ability in the human auditory system that allows us to converse in noisy environments. As long as the solar panel or photodiode at the receiving end has a shield so that it is not disturbed by ambient light, it can ensure that the user only hears the conversation. In signal processing, because the voice transmission is through modulations in the laser light, we can use low-computing digital filtering algorithms running on a low-cost DSP to reduce design costs. Advanced bone conduction hearing aids can help to avoid the cocktail party problem. It is necessary to use multiple microphones with an advanced beamforming algorithm to remove spatial noise, making the design more complex. The cost is higher than that of general bone conduction hearing aids. Therefore, our research provides a low-cost and easy-to-design dentary bone conduction system framework for optical communication to transmit voice with the goal of being a new research field in bone conduction in the future. The performance comparison of voice signal wireless transmission methods is shown in Table 1. We also compare the performance difference between our dentary bone conduction device and other bone conduction devices, which is shown in Table 2.

## 2. Materials and Methods

Figure 3 and Figure 4 show the PCB layouts of the laser transmission device and the receiving device, respectively. They were designed by us with open-source CAD software (Eagle PCB). The laser audio transmission device (Figure 5) can receive audio from a microphone or use a 3.5-mm audio cable to receive analog audio. Figure 5a displays the switch: up is open, down is closed. Figure 5b shows the voltage switch, set to a voltage of 4.2 V, with the jump changed to 5 V (not recommended). Figure 5c displays the audio input: one may use a 3.5-mm earphone plug to input the computer audio, and the laser will output the audio signal. Figure 5d shows the audio output: one can confirm that the input audio is used normally and directly connects to a 3.5-mm earphone. Figure 5e displays the switch between the microphone and audio input: it can be set to the left for audio input or to the right for microphone input. The two audio sources are modulated into the laser. The AC signal of the audio source can change the output of the amplifier, causing the laser light intensity to have an AC signal so that the target receiver pointed at by the laser can transmit audio information. We also designed a laser audio signal receiver (Figure 6) to receive the laser signal from the transmission device through a solar panel or PD and convert the photoelectric signal into an analog signal through a differential amplifier [15]. Figure 6a shows the PAM8403 output earphone, which can drive bone conduction. Figure 6b shows the trans-impedance amplifier (TIA) output earphone, which can be used to listen to the sound transmitted by the solar panel or photodiode. Figure 6c displays the power switch, which turns on and off. Figure 6d displays the input of the piezoelectric sensor. The output signal of the differential amplifier can be input to a 3-watt D-class power amplifier. The power amplifier can directly drive the BCT or input the 12-bit analog into the microprocessor’s digital converter (ADC).

The digital filtering algorithm running on the microprocessor processes the audio. The signal is processed before it is input to the power amplifier so that the BCT can output a cleaner sound signal to the user. Figure 7 shows the system architecture. The transmission of audio by the laser can differ from mechanical wave transmission, and it is not easily affected by the physical environment. Moreover, the laser light intensity at a distance of a few meters is relatively small, which can be beneficial for the transmitted audio. The signal-to-noise ratio makes it easier to improve the BCT’s audio quality through signal processing.

### 2.1. Laser Audio Transmitter and Receiver

The laser audio transmission device can modulate the audio signal to the laser through the LM358, a chip that encapsulates two operational amplifiers. The device can also connect to the audio playback device through the 3.5-mm audio cable, and the audio can be input into the positive terminal of the amplifier (Figure 8). Therefore, the amplifier’s output voltage will output the same amplitude as the AC signal of the input audio, and the input of the negative terminal determines the other part of the output voltage. The DC level of the amplifier’s output voltage can be adjusted by the negative terminal feedback resistor’s potentiometer. We also designed a microphone amplifier circuit (Figure 9) that can receive the microphone’s input voice signal to facilitate communication between healthy people and patients with ruptured eardrums. The microphone connects to a pull-up resistor audio filter using a high-pass filter composed of a 1 uF capacitor and a 10 kΩ resistor. The high-pass filter’s cutoff frequency is 15.9 Hz, which covers the lowest human hearing frequency (20 Hz). After high-pass filtering, the output signal is input to the negative terminal of the amplifier. The amplifier output voltage will change according to the microphone’s audio amplitude and frequency with the positive terminal input resistance and the amplifier voltage divider’s negative resistance. The amplifier’s output can modulate voice signals to the laser. Two types of audio source inputs can be selected by switching. In this device’s design, the laser diode power used was a 3 to 5 milliwatt red laser with a wavelength of 650 nm.

This study uses an 8 × 8 cm solar panel with an output voltage of 5 V and a current of 100 mA or another PD that can receive a 650-nanometer laser signal response. When the laser irradiates the solar panel, the solar panel can passively generate a current. The laser audio transmission device can modulate the AC signal of the audio to the laser’s intensity, so the current generated by the solar panel will also change along with the AC signal, and the audio signal generated by the solar panel will be input into the LM324. LM324 has four rail-to-rail amplifiers, and the amplifier that receives the solar panel’s current is a TIA (Figure 10). The TIA can effectively convert the current into voltage for the PD that generates a current due to the photoelectric effect. The TIA’s output voltage can be input into a D-class amplifier to drive the BCT [16] or a 12-bit ADC for signal processing. The D-class amplifier is a PAM8403 amplifier. The internal input resistance, Ri, of PAM8403 is 18 kΩ, and the feedback resistance is 142 kΩ. The experimental setting has an external input resistance of 10 kΩ, and the close-gain calculation formula is 20 × log (2 × (142 kΩ/[18 kΩ + 10 kΩ])), which equals 31 dB [17]. PAM8403 can amplify the TIA’s voltage output and drive the BCT, and the BCT can convert the audio AC signals into mechanical waves. The BCT can transmit audio to the acoustic nerve as mechanical waves when it contacts the dentary bone.

This study uses piezoelectric ceramics to receive the mechanical wave generated by the BCT to measure the feedback signal [18]. The mechanical wave will cause the piezoelectric material to undergo a piezoelectric effect so that the piezoelectric ceramic can output mV-level voltage. For this analog front-end (AFE) design, we used three amplifiers of LM324, pulled down the positive pole of the piezoelectric ceramic to a 4.7 mΩ resistor, and input the positive terminal (Figure 11) of the first-stage voltage follower. The voltage follower’s input resistance is infinite, and the zero output resistances can ensure the complete input of tiny piezoelectric signals. The second-stage amplifier is a feed-forward amplifier (Figure 12) with a 19-times gain. It can amplify the voltage follower’s mV-level signal output to increase the signal-to-noise ratio (SNR). The last-stage amplifier is a low-pass filter (Figure 13) with a Sallen–Key structure. The cutoff frequency is 72 kHz. Although the feed-forward amplifier’s output amplifies the audio’s AC signal, it also amplifies the noise. Therefore, we designed the low-pass filter to ensure that the upper limit of the frequency covering human hearing is 44.1 kHz so that this design is in line with the frequency range we want to analyze. The low-pass filter output sampled a 12-bit ADC at a frequency of 44.1 kHz and was analyzed using a digital signal processing technique.

We used an STM32F407 processor [19] integrated with a 12-bit ADC to perform laser audio sampling (Figure 14). STM32F407 can transmit audio to a computer in digital packets using the USB audio class protocol of USB2.0 for analysis. This function is similar to that of a sound card. We receive the TIA’s output for microphone recording to analyze the audio quality transmitted by the laser. We also received the low-pass filter’s output to analyze the mechanical wave output by the BCT directly measured by the piezoelectric ceramic. We then analyzed whether the output of the BCT converted from electrical signals into mechanical waves was distorted. We will discuss the results in Section 4.

### 2.2. Analog Front End

This section explains how we set up an analog-to-digital conversion system for receiving laser audio analog signals. After receiving the audio from the transmitter, the laser audio receiver outputted the analog voltage signal via the TIA. We then used the integrated 12-bit SAR-ADC ARM SoC to receive analog signals and converted them into quantized digital signals.

#### 2.2.1. Analog to Digital Converter (ADC)

The STM32F407 processor is a platform for digital signal processing. The processor integrates a 12-bit SAR-ADC [20], which can quantize audio from 0 to 3.3 (analog voltage) to 0 to 4095 (digital signal). To cover the audio frequency range perceptible to the human auditory nerve, we set the processor to have a sampling rate of 44,100 Hz for the ADC. We set the timer interrupt period of the processor to 1/44,100 s. When the timer interrupt was triggered, direct memory access was triggered to sample the ADC’s analog signal and convert it to a digital value of 0 to 4095. The flowchart is shown in Figure 15. The STM32F407 pins PC5 and PC4 connect to the TIA output and low-pass filter output of the laser audio receiver. It can convert analog signals into digital signals, thereby providing a feasible means for laser audio signal processing.

#### 2.2.2. Digital-to-Analog Converter (DAC)

After the laser audio converts the analog signal to a digital signal, we can use digital signal processing algorithms to filter the sound signal, thereby improving the audio quality or strengthening the audio frequency band that we wish users to hear. It must transform from a digital signal to an analog signal after processing to output the digital signal to the BCT. We used the 12-bit DAC [21] integrated inside the STM32F407 to convert the digital signal processed by the digital filtering algorithm from 0 to 4095, to 0 to 3.3 analog voltage. Furthermore, we had to output to the D-class amplifier to drive the BCT to enhance the audio output. The connection method for DAC and the D-class amplifier is shown in Figure 16.

## 3. DSP Algorithm Design

This section explains the digital signal processing method of laser audio signals. To improve the audio quality output by the BCT, we used STM32F407 and a 12-bit ADC to sample the TIA amplifier’s work at the audio receiver. The filter algorithm runs in the STM32F407 processor to filter the laser audio signal before the DAC outputs the filtered signal. A finite impulse response (FIR) filter is a stable digital filtering algorithm, and the filter coefficients can be designed through simulation software. Using the frequency response and window function designed by simulation software [22], the coefficient can be generated after the product.

Furthermore, the time-domain signal and the coefficient are convoluted (Equation (1)) so that the filtered time-domain signal can be obtained. The FIR filter algorithm can run on the STM32F407 processor in real time because of its stability and limited length. After the received laser audio is processed in real time, the DAC is output to the PAM8403 to drive the BCT to produce clearer audio.
(1)$$y[n]=\sum_{k=0}^{N-1}b_{k}\cdot x[n-k]$$
where x[n] is the signal before filtering, y[n] is the filtered signal, $b_{k}$ is a filter coefficient.

To design an FIR filter, we determined the signal sampling rate and filter cutoff frequency. We calculated the frequency response and window function (the coefficients) according to different FIR filter types and filter lengths. We used a band-pass filter (Equation (2)) for frequency response and set the cutoff frequency *f*_*c*1_ to 200 Hz and *f*_*c*2_ to 3000 Hz. This range covers the frequency range of general adult speech [23,24]; *ω*_*c*1_ and *ω*_*c*2_ are band-pass filter cutoff frequencies, *f*_*c*1_ and *f*_*c*2_, converted to the transition frequency [25], which needs to be divided by the sampling rate of the ADC 44,100 Hz (Equation (3)). We used the hamming window (Equation (4)) for the window function, and the *a*0 and *a*1 parameters were set to 0.54 and 0.46, respectively [26]. Then, we used the product window functions and frequency response (Equation (5)) to obtain the filter coefficient *b*[*k*]. Finally, we convolved the input time-domain signal *x*[*n*] and the filter coefficient *b*[*k*] to obtain the filtered signal *y*[*n*].
(2)$$h_{d}[n]=\{\begin{array}{c} \frac{\sin(\omega_{c2}(n-N))}{\pi (n-N)}-\frac{\sin(\omega_{c1}(n-N))}{\pi (n-N)};n\neq N\\ \frac{\omega_{c2}-\omega_{c1}}{\pi };n=\frac{N}{2} \end{array}$$
(3)$$\omega_{c1}=f_{c1}\frac{1}{f_{s}};\omega_{c2}=f_{c2}\frac{1}{f_{s}};f_{s}=44,100\;\mathrm{Hz}$$
(4)$$w[n]=0.54-0.46\cos(\frac{2\pi n}{N});0\leq n\leq N$$
(5)$$b[n]=w[n]\cdot h_{d}[n];0\leq n\leq N-1$$

## 4. Result and Analysis

In this section, we analyze the output signal of the TIA of the laser audio receiver device and the signal output after DSP filtering and compare the original signal of the LM358 amplifier of the laser audio transmitter. After the signal was normalized, we used the signal-to-noise ratio (*SNR*) (Equation (6)) function to analyze the difference between the audio quality transmitted by the laser and the audio quality filtered by the DSP.
(6)$$SNR=20\log10(\frac{S}{N})$$
where *S* is the amplitude of the raw signal before the laser transmission and *N* is the received noise of the raw signal.

### Result Analysis

We used the voice in the CMU ARCTIC database as the standard audio signal for laser audio transmission in this study. This audio set is an open-source database [27] recorded by the Language Technologies Institute at Carnegie Mellon University for language technologies research. We selected the arctic_a0223.wav sequence from the cmu_us_awb_arctic group in the database as the standard audio source.

Our experiment is in a meeting room with fluorescent lights (60 Hz frequency as the periodicity of the mains electricity supply) on. We use the laser audio transmitter source as the signal source for SNR calculation. Because there is a time difference between the signal transmitted by the laser and the signal at the receiver, the digital filtering of the DSP will cause a phase shift. We process the signal from the transmitter and the receiver with the alignsignals method of MATLAB to ensure the consistency of the SNR calculation. For the BCT output signal, the same method was used to process the SNR calculation and compare the SNR between the BCT output signal that uses DSP with digital filtering and the unfiltered BCT output signal.

Figure 17 shows the output signal of the TIA and the signal filtered by DSP. The upper signal of Figure 17 is unfiltered. The audio output by the TIA sounds like high frequency and noise to the human ear, and it is impossible to hear the reading content of the standard audio source. The bandpass filter filtered the lower signal of Figure 17, and a signal exceeding 100 Hz to 10 kHz was filtered out. The high-frequency noise reduced the filtered signal, and the SNR was improved compared with that before and after filtering. The DSP output audio can represent the reading content in the standard audio source.

Figure 18 shows that the piezoelectric sensor received the vibration signal of the BCT. The upper panel of Figure 18 shows that the TIA signal directly outputted the BCT’s vibration signal without being filtered by the DSP. The lower panel of Figure 18 shows the output of the bone conduction after being filtered by the DSP. It shows that the vibration signal of the BCT output by the DSP is cleaner and noise free. The vibration signal of the BCT output by the DSP also has a higher SNR compared to the vibration signal of the BCT output by the TIA. It is consistent with the SNR trend in Figure 17.

Through the measurement of the vibration signal of the BCT by the piezoelectric sensor, we verified that using a low-cost DSP with a digital filtering algorithm can increase the SNR from 13 to 21 dB. One can also observe, in Figure 18, that even if DSP filters output the vibration signal of the BCT, there will be some noise interference. We believe that the reasons are twofold. One reason points to the assembly method and mechanism design of the BCT. The other is attributed to the disturbance when the piezoelectric sensor receives the BCT’s vibration signal measurement. In the future, it will be necessary to accurately measure the vibration signal of the BCT to use the closed-loop control algorithm on the DSP to control the vibration of the BCT more accurately. Researchers of dentary bone conduction devices must pay attention to these two issues.

## 5. Conclusions and Future Work

This article proposes a system that can transmit audio by laser and use DSP to output filtered audio vibration signals to dentary bone. We designed a laser audio transmitter that can receive audio from audio playback equipment or microphones. A receiver can obtain a laser audio signal through a PD or solar panel and drive a BCT. It also implements a DSP filter that processes the laser audio signal received by the receiver. Furthermore, the digital filter can filter out the audio’s noise in real time and then output it to the BCT. In our experimental results of transmitting standard audio sources read by adults, the SNR of the audio signal transmitted by the laser after being filtered by DSP was higher than that before being filtered. The SNR of the vibration signal output by the BCT also showed the same trend. The audio received by the user through the dentary bone was markedly clearer. We also implemented the method of measuring the BCT’s vibration signal through a piezoelectric sensor and provided the measurement of the BCT’s actual output, which is the basis for realizing the closed-loop control of the output compensation of the BCT on the DSP. In the future, based on the system proposed in this research, we hope that by measuring the vibration signal of the BCT, we can develop more bone conduction digital signal processing algorithms to achieve compensation for output distortion and improve user hearing. To develop the industry, a dental bone conduction device that uses a laser to communicate would be a valuable means of communication in space where there is no gas (medium). As such, this article should be interpreted, and used as, a pilot study. Because teeth are harder than the skin, they can conduct audio more clearly than the skin on the skull. Moreover, in the sea or in outer-space environments, optical communication using lasers can be used as a method of communication. If it is in the field of the Internet of Things, this device can also be used to broadcast. Because RF is easily interfered with by environmental electromagnetic waves, when the user wears a bone conduction hearing aid with RF signal, it may receive sound from all directions, but when there may be too many sound sources and continuous noise, the user may not be able to correctly receive what they want to hear.

## Figures and Tables

**Figure 1 sensors-21-04229-f001:**
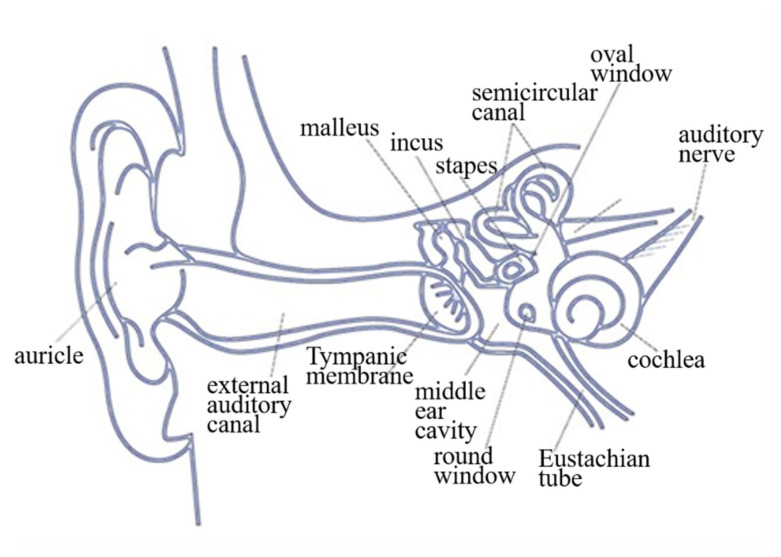
Structure of the human ear.

**Figure 2 sensors-21-04229-f002:**
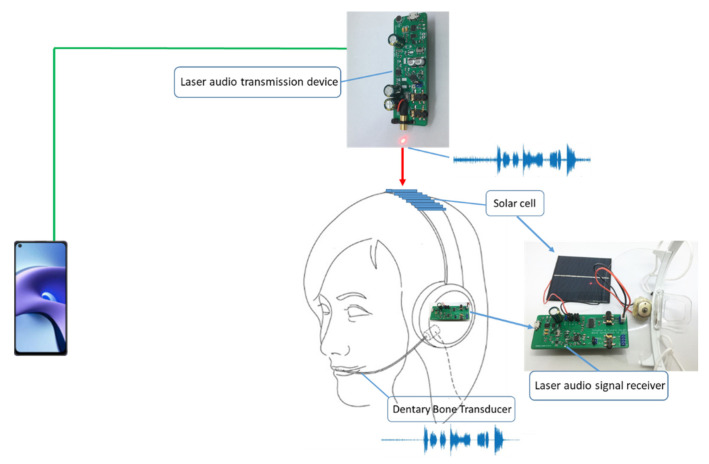
The schematic diagram of a dentary bone conduction system applied to humans.

**Figure 3 sensors-21-04229-f003:**
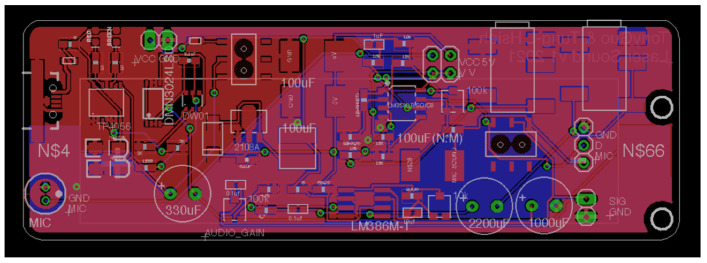
The PCB layout of the laser transmission device.

**Figure 4 sensors-21-04229-f004:**
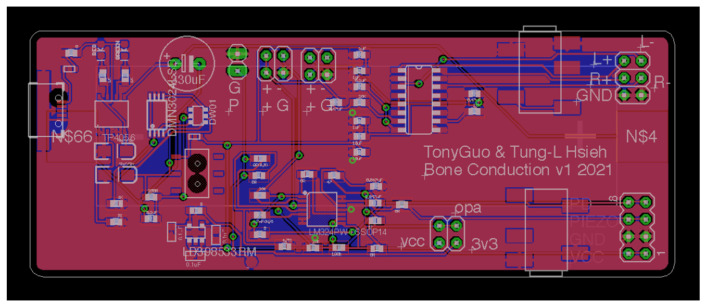
The PCB layout of the laser receiving device.

**Figure 5 sensors-21-04229-f005:**
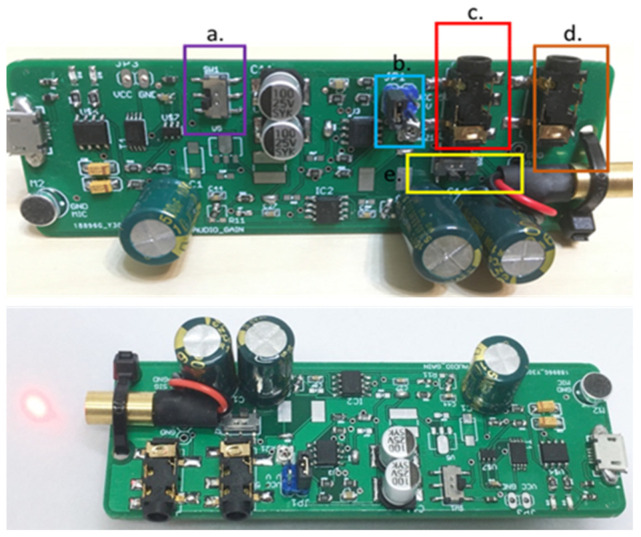
Laser audio transmission device. a displays the switch: up is open, down is closed. b shows the voltage switch, set to a voltage of 4.2 V, with the jump changed to 5 V (not recommended). c displays the audio input: one may use a 3.5-mm earphone plug to input the computer audio, and the laser will output the audio signal. d shows the audio output: one can confirm that the input audio is used normally and directly connects to a 3.5-mm earphone. e displays the switch between the microphone and audio input: it can be set to the left for audio input or to the right for microphone input.

**Figure 6 sensors-21-04229-f006:**
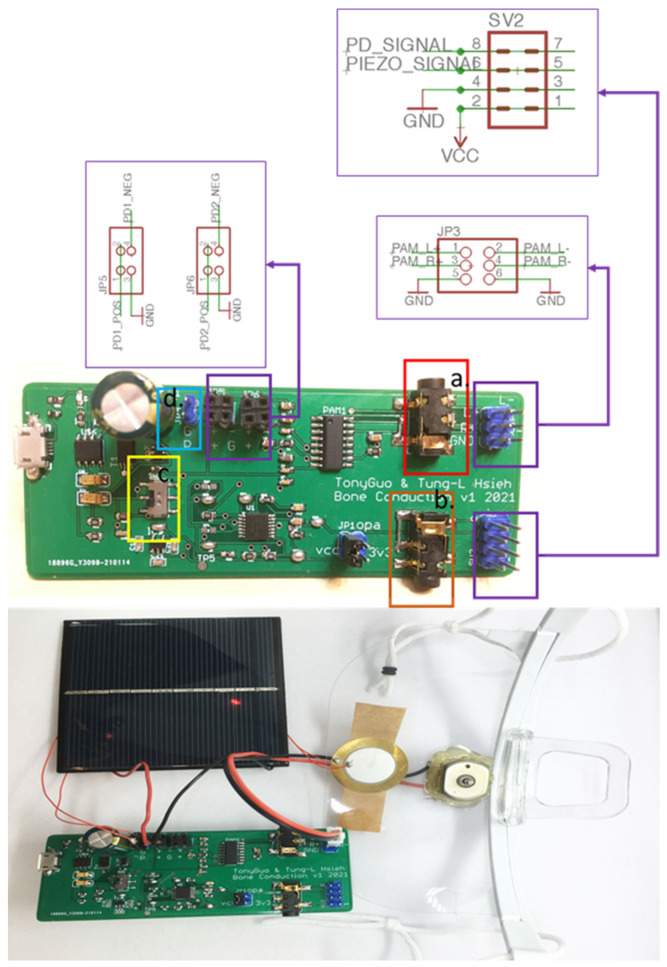
Laser audio signal receiver. a shows the PAM8403 output earphone, which can drive bone conduction. b shows the trans-impedance amplifier (TIA) output earphone, which can be used to listen to the sound transmitted by the solar panel or photodiode. c displays the power switch, which turns on and off. d displays the input of the piezoelectric sensor.

**Figure 7 sensors-21-04229-f007:**
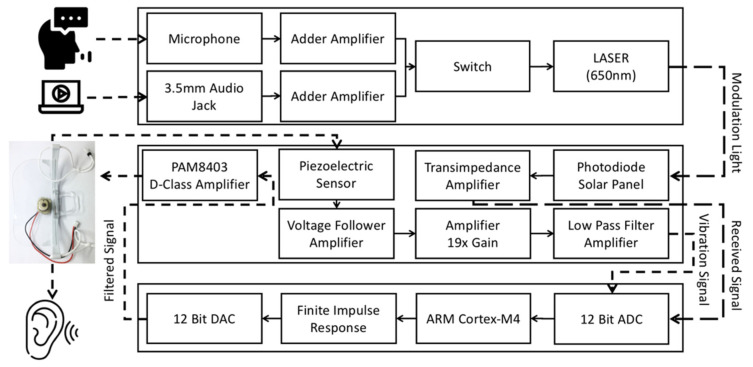
System architecture.

**Figure 8 sensors-21-04229-f008:**
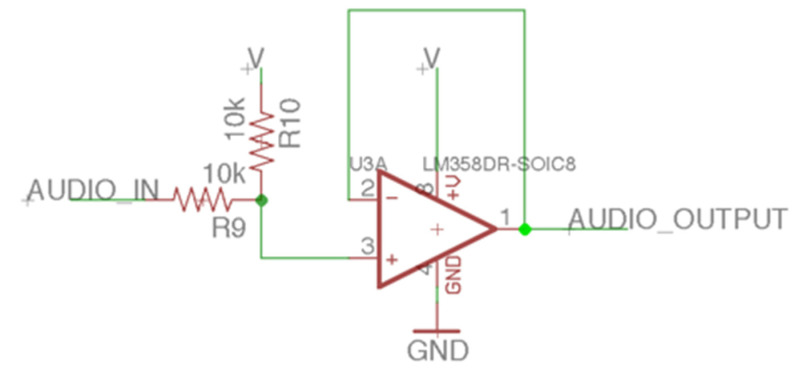
The audio signal input circuit of the laser audio transmission device.

**Figure 9 sensors-21-04229-f009:**
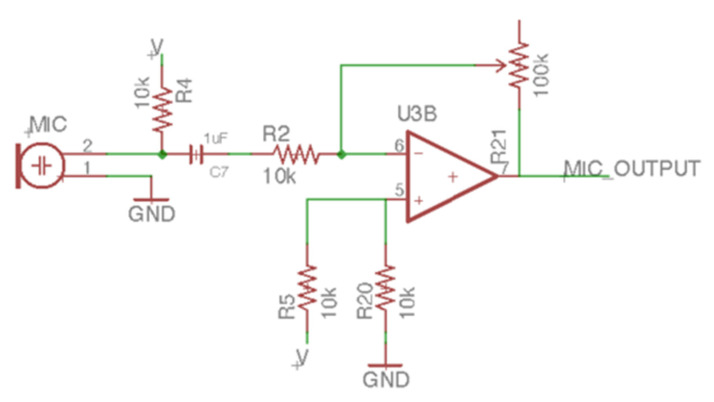
The microphone circuit of the laser audio transmission device.

**Figure 10 sensors-21-04229-f010:**
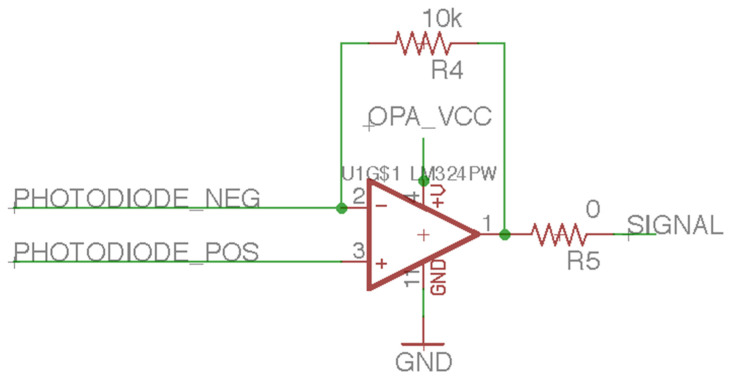
The trans-impedance amplifier of the laser audio signal receiver circuit.

**Figure 11 sensors-21-04229-f011:**
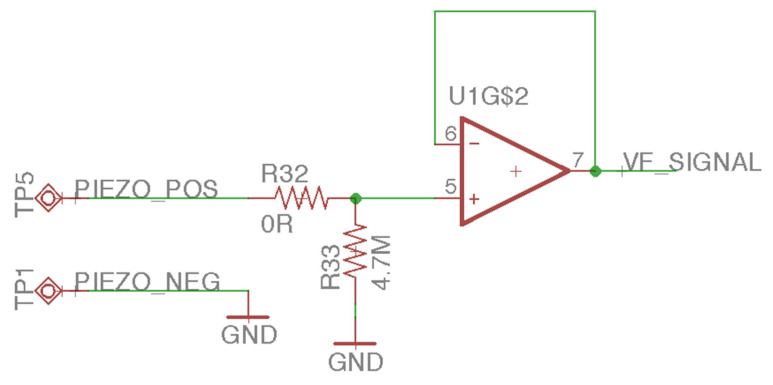
The voltage follower of the laser audio signal receiver circuit.

**Figure 12 sensors-21-04229-f012:**
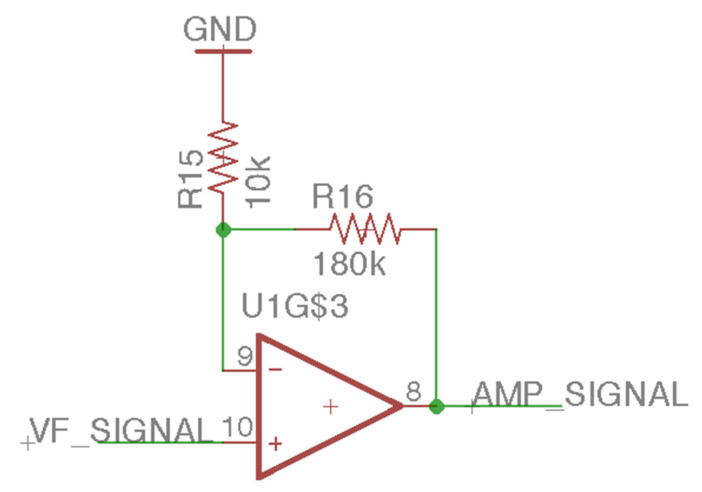
The feed-forward amplifier of the laser audio signal receiver circuit.

**Figure 13 sensors-21-04229-f013:**
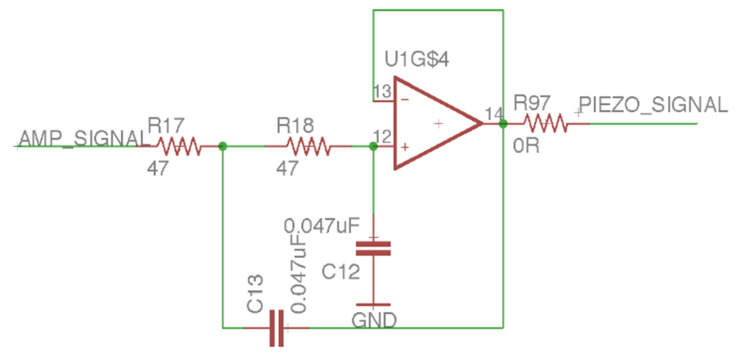
The low-pass filter with a Sallen–Key type of laser audio signal receiver circuit.

**Figure 14 sensors-21-04229-f014:**
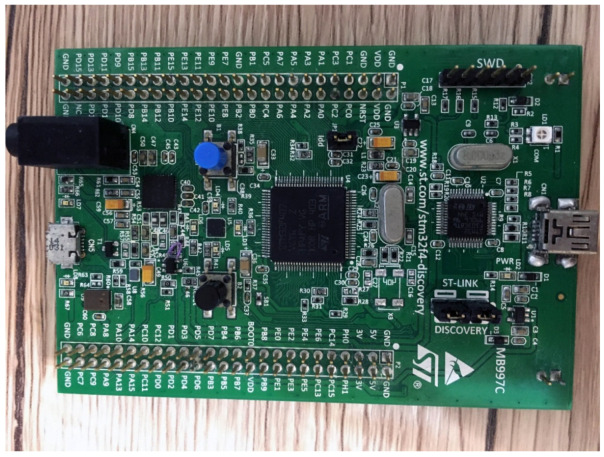
STM32F4-Discovery and 12-bit ADC.

**Figure 15 sensors-21-04229-f015:**
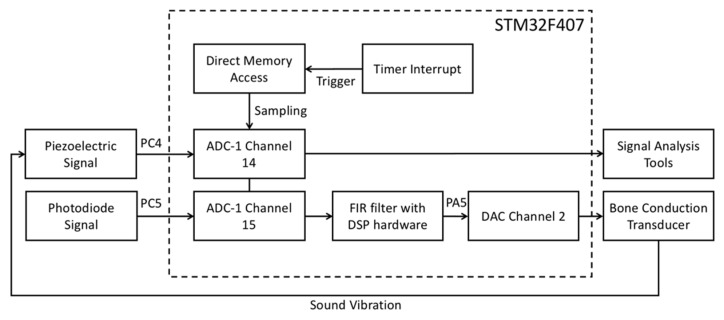
Digital signal processing flowchart.

**Figure 16 sensors-21-04229-f016:**
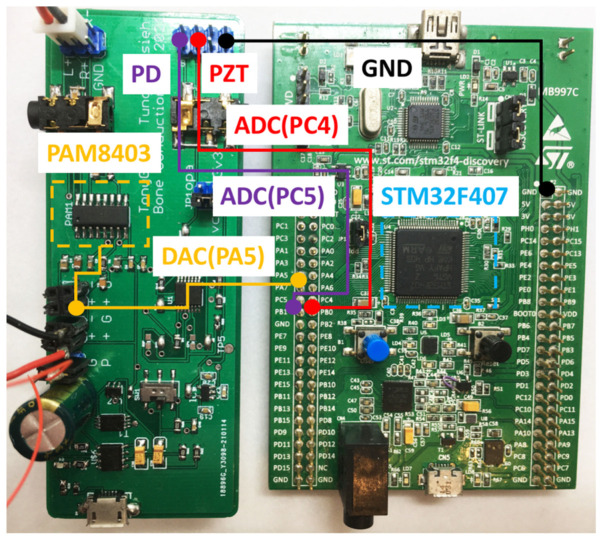
Connection between the laser audio receiver and the STM32F407 processor.

**Figure 17 sensors-21-04229-f017:**
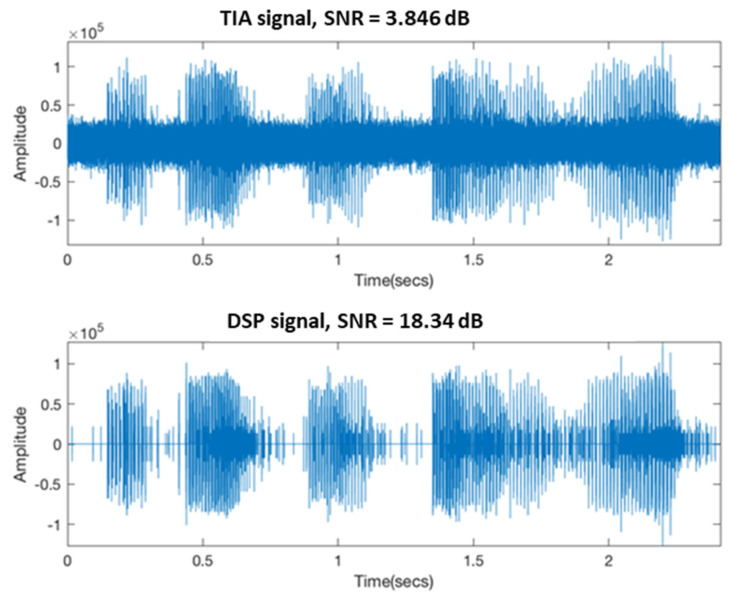
Time−domain signal of TIA and DSP output.

**Figure 18 sensors-21-04229-f018:**
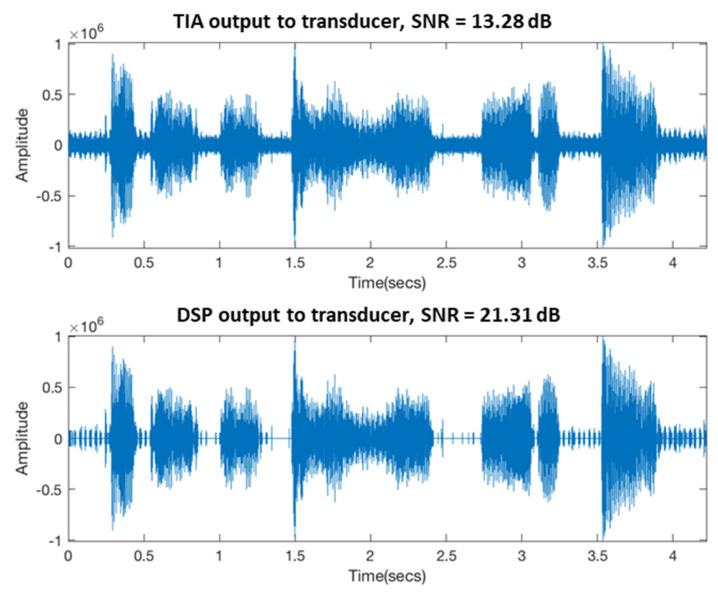
Time−domain signal of vibration of BCT.

**Table 1 sensors-21-04229-t001:** The performance comparison of voice signal wireless transmission method.

	Laser	RF
Radio frequency interference	No	Yes
Transport security	High (cannot be sniffed)	Low (can be sniffed)
The computational complexity of signal processing algorithms	Simple (digital filter)	Complex (decoding algorithms, arithmetic coding, error-correcting code, FFT, digital filter)
Cost of analog front end	Low (transimpedance amplifier, gain amplifier)	High (low-noise amplifier, voltage-controlled oscillator, mixer)

**Table 2 sensors-21-04229-t002:** The performance comparison between our dentary bone conduction device and other bone conduction devices.

	Dentary Bone Conduction	Other Bone Conduction
Voice signal attenuation by soft tissue	No	Yes
Cocktail party effect	No	Yes (background noise coupled to the microphone)
Robust	Yes	No (head-mounted displacement caused by movement)

## Data Availability

Not applicable.

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
