# Peer review of "A Novel Dentary Bone Conduction Device Equipped with Laser Communication in DSP"

_sensors, 2021, doi:10.3390/s21124229_

Round 1

Reviewer 1 Report

While the idea is generally interesting, there are aspects that the authors need to clarify. In addition, the presentation, and the level of English (several sentences are difficult to read and/or convoluted and there are problems with the figure numbering) need to be improved. The scenario in which the proposed device is intended to be used is not clear to determine whether laser systems are a suitable choice. In the introduction, the authors say that the proposed laser-based communication system improves usability by “preventing audio signal attenuation”. Laser-based communication can reduce losses (not prevent them, since even a collimated Gaussian beam suffers spatial dispersion) but at the same time introduces new problems such as the line-of-sight requirement. Authors should clarify more strongly what the actual usage scenario is and comment on the issues of shadowing, signal interruption, displacement/rotation of the user within the scenario with respect to the transmitting system. Compared to that scenario, what advantages/disadvantages has the laser communication system over an RF system? Further, the authors state they use a 3W to 5W !!! red laser diode @ 650 nm, how do they contextualize such values with respect to safety regulations?

Reviewer 2 Report

I could not evaluate that the new device itself you have developed has a novel point from the viewpoint of hardware technologies. To show this, you should additionally show the profits by actually using your device as some human experiment. In contrast, from the viewpoint of acoustics technology, I evaluated that this paper has effective and novel points after chapter 3. But, in chapter 3, I could only see just filtered voice waveforms that are difficult to sufficiently show the above novelty and effectiveness. It can be read as you only filtered out the noise pollution by the laser transmission. From the viewpoint of acoustics, the filtering scheme indicated in chapter 3 is quite an ordinary signal processing, and the scheme itself is not new. Besides, the results of Figures 16-19 are quite easily imaginable outcomes. If I must say, the mechanism of the noise pollution by laser transmission is one of the effective points, but it is not detailedly treated. Besides, your title has the word of bone conduction, however, I could find out how the bone conduction relates to the novelty and effectiveness of this device technology. So, I suggest rejection, however with a major revision of the above, I think resubmission will be encouraged.

Title
The wording of self-made laser communication is not appropriate, since it is not important whether it is self-made or not.

Introduction
Please clearly describe in the last paragraph of the introduction what you did and you clarified. In the present state, it is difficult to briefly understand what was done in this paper.

English quality
The English should be grammatically and editorially proofread by professional service.

e.g.) 
"Furthermore, the time-domain signal and the coefficient are convolutions (eq.1) that the filtered time-domain signal can obtain."

=> At least "convolutions" should be "convoluted" .

"where y[n] is filtered, signal b_k is filter coefficient, and x[n] is signal before filter."
=>
"where x[n] is signal before filtering, y[n] is filtered signal,  b_k is filter coefficient."

"Spectrum-analysis" should be "Spectrum analysis" without a hyphen.

"The method we use is to perform a short-time Fourier transform (STFT) on unfiltered and filtered signals."
=> A short-time Fourier transform (STFT) was used...

"noise-contribution"
=> Your way of hyphenation should be re-reviewed.

"Figure 18 shows that the piezoelectric sensor receives the vibration signal of the BCT."
=>Figure 18 shows the vibration signal of BCT received by the piezo sensor. 

Eq. (6)
S is not the "raw signal" but "the amplitude of" the raw signal. Please re-review throughout your paper to keep an acoustically accurate description.

Figure 16

>> Please clearly show how you calculated SNR of 2.08 and 5.896.
>> Please use the appropriate significant digits for these numbers.
>> What is the quantity of the horizontal axis?

Noise effect
Related to the noise effect caused by laser transmission, I could only confirm one sentence below. The sound wave of the transmitted voice is quite polluted by pulsive but continuous noises as shown in Fig. 16. (It seems the noise has quite broad frequency components) But, in this paper, the frequency characteristics of the noise are not detailedly and quantitatively discussed. Please show the frequency response of the noise. STFT is not a good way to quantitatively know the frequency response of the noise. Besides, I'd also like to know that the frequency characteristics of noise pollution in many cases are almost the same or quite changeable. It largely relates to the generality of the investigated contents.

"The noise-contribution is a result of the impact of ambient light on the PD or solar panel during the laser's audio transmission."

Figure 18
>>Both of these signals have negative values of SNR, but it can't be understood these signals have SNRs not with positive but with negative values. I'd like to strongly know how you calculated SNRs.
>> What is the quantity of the horizontal axis?

Reviewer 3 Report

The authors may please outline in bullets in the introduction what are the novelty points of this document either from the device side or from the signal processing side. 

Also, the authors comment that the theme of this work is "rare" and give [13] as a justification; yet [13] is a work in 2014. Are the authors sure that no subsequent works have emerged?

From the devices side, the authors must mention what are the novel designs the authors have considered over existing works (I am sure the laser-based bone conduction system; in case it is so, they must pinpoint the novelty in the introduction as bullet points).

From the signal processing side, the novelty is again not clear. The pointwise highlight of novelty in the form of bullet-points in the introduction is so important; then you can give the verbose description of your implementation, no problems there.

Also, the paper starts so abruptly; "The sound is transmitted through bones"!!! Which sound and whose bones? This is an example of the fact that further proofreading is needed to make the contribution clearer.

Though I am sure that the authors mean well, they may perhaps pinpoint the distinguishing features of their implementation over existing works and it is more than fair to hear their arguments in this regard and read the clarifications in their response.

I look forward to receiving their revision.

Round 2

Reviewer 2 Report

Accepted.

Reviewer 3 Report

The authors have made it much clearer now. It can now be accepted.